# Stretch-Induced Tenomodulin Expression Promotes Tenocyte Migration via F-Actin and Chromatin Remodeling

**DOI:** 10.3390/ijms22094928

**Published:** 2021-05-06

**Authors:** Pu Xu, Bin Deng, Bingyu Zhang, Qing Luo, Guanbin Song

**Affiliations:** 1Key Laboratory of Biorheological Science and Technology, Ministry of Education, College of Bioengineering, Chongqing University, Chongqing 400044, China; xupu@cqu.edu.cn (P.X.); dengbin09ke4@163.com (B.D.); qing.luo@cqu.edu.cn (Q.L.); 2Chongqing Engineering Research Center of Medical Electronics and Information Technology, College of Bioinformatics, Chongqing University of Posts and Telecommunications, Chongqing 400065, China; zhangbinyu1111@126.com

**Keywords:** tenomodulin, tenocyte migration, mechanical stretching, actin stress fibers, chromatin remodeling

## Abstract

The mechanosensitive gene tenomodulin (Tnmd) is implicated in tendon maturation and repair. However, the mechanism by which mechanical loading regulates Tnmd’s expression and its role in tenocyte migration is yet to be defined. Here, we show that Tnmd and migration were upregulated in uniaxial cyclic stress-stimulated tenocytes. The knockdown of Tnmd reduced cell migration in the presence and absence of mechanical loading, suggesting that Tnmd is involved in tenocyte migration. Moreover, the treatment of stress-stimulated tenocytes with the actin inhibitor latrunculin (Lat A), histone acetyltransferase inhibitor anacardic acid (ANA), or histone demethylases inhibitor GSK-J4 suppressed Tnmd expression and tenocyte migration. These results show that actin stress fiber formation and chromatin decondensation regulates Tnmd expression, which might then regulate tenocyte migration. Thus, this study proposes the involvement of the actin and chromatin mechanotransduction pathway in the regulation of Tnmd and reveals a novel role of Tnmd in tenocyte migration. The identification of Tnmd function in tenocyte migration provides insight into the molecular mechanisms involved in Tnmd-mediated tendon repair.

## 1. Introduction

Tendons are mechanosensitive tissues with a highly organized collagen structure that connect muscle to bone and function as force transmitters during movement [1]. Consequently, tendons withstand tremendous physiological loading, and inappropriate physical training or excessive repetitive stretch can lead to tendon injuries [2]. Tenocytes, the main functional cells of tendons, are embedded in the collagen fiber bundles and responsible for maintaining tendon homeostasis and biomechanical properties through sensing and responding to biomechanical signals [1,3]. Upon tendon injury, tenocytes migrate to the damaged site and facilitate the repair process through the secretion of ECM, thereby making them key players in the tendon tissue regeneration and repair process [4]. Therefore, the molecular mechanisms that govern tenocyte migration are essential in understanding the tendon repair process. 

Tenomodulin (Tnmd), a type II transmembrane glycoprotein, has been identified as a tendon maturation-specific marker that plays an essential role in stem/progenitor cell (TSPC) proliferation, differentiation and senescence [5,6]. Moreover, the absence of Tnmd level in Tnmd knockout (Tnmd-/-) *mice* reduced tenocyte proliferation and induced the pathological thickening of collagen fibers in the tendon extracellular matrix (ECM), which resulted in significantly inferior functional performance [7,8]. Tnmd-deficient TSPCs have also been shown to exhibit severe migratory deficit [7]. On the contrary, Tnmd overexpression enhanced tendon-like tissue formation in vivo and positively correlated with tendon healing advancement [9]. These results suggest that maintained Tnmd expression in tendons might be necessary for tenocyte migration during tendon repair. Moreover, Tnmd has been identified as a mechanosensitive gene induced in developing and adult tendons, and its expression in tendons correlates with force intensity [10,11]. Despite the proposed mechanosensitive property of Tnmd and its role in migration, little is known about the mechanisms that induce Tnmd expression in the tendon biomechanical environment. 

Generally, cells within the tissue perceive the biomechanical signals and respond by activating various mechanotransduction signaling pathways that result in the production of molecular factor-involved ECM remodeling and tissue repair. The perception and translation of biomechanical signals in the cytoplasm are highly dependent on the cell cytoskeleton [12]. Especially, the actin cytoskeleton plays a mechanoregulatory role in producing a coordinated and directed cellular response to mechanical stimuli [13]. Notably, the actin cytoskeleton responds by forming contractile actomyosin bundles, stress fibers that convert extracellular stimuli into intracellular signals [14]. Forces generated by extracellular strain transmitted through the actin stress fibers to the nucleus, resulting in transcriptional activity modulation through different mechanisms. One of the potential mechanisms by which transduced forces alter gene transcription in the nucleus is chromatin physical organization modulation [15]. Mechanical forces modulate chromatin condensation, which in turn leads to the differential accessibility of DNA to regulate gene transcription. Moreover, it has been suggested that disruption of the actin stress cytoskeleton abolishes the force-mediated transcription response [15]. Although these results indicated the involvement of actin cytoskeleton-mediated chromatin organization in force-induced transcription, whether this pathway is involved in regulating biomechanically induced Tnmd expression is yet to be demonstrated.

This study investigates the effects of cyclic mechanical stretching on Tnmd expression and uncovers the role of Tnmd in the stretch-induced tenocyte migration. We also explored the potential mechanotransduction mechanism that results in Tnmd expression and tenocyte migration, focusing on actin cytoskeleton-mediated chromatin organization. Understanding how tenocytes sense mechanical stimuli and ultimately translate them into specific biological outcomes is essential for advancing the field and developing new clinical therapies for injured tendons.

## 2. Results

### 2.1. Mechanical Stretching Induces Tnmd Expression and Cell Migration in Cultured Tenocytes

To confirm the expression of Tnmd in tendon tissue and cells, the mRNA and protein analysis of Tnmd confirmed that Tnmd was markedly reduced in cultured tenocytes compared with tendon tissue (Figure 1A,B). Next, the gene and protein expression of Tnmd was analyzed in loaded tenocytes. The results showed that tenocytes exposed to stretching of 10% strain, 0.5 Hz, for 1 h had a higher expression of *Tnmd* at the mRNA (Figure 1C) and protein (Figure 1D,E) levels compared to the unstretched tenocytes. Moreover, the transwell assay was utilized to detect the migration behavior of mechanically loaded tenocytes. Data revealed a significant increase in tenocyte migration in loaded tenocytes (Figure 1F,G). These results infer a possible association between mechano-induced Tnmd expression and tenocyte migration.

### 2.2. Stretching-Increased Tnmd Regulates the Migration of Tenocytes

To further investigate whether Tnmd could directly regulate the tenocyte migration, the migration ability of tenocytes was assayed after Tnmd knockdown with lentiviral short hairpin RNA (sh-RNA). Compared with the control silencing group (pLKO.1), the mRNA and protein of Tnmd in Tnmd-knockdown groups (sh-Tnmd1 and sh-Tnmd2) decreased significantly (Figure 2A,B). Moreover, stretch-stimulated tenocytes showed a higher Tnmd expression than Tnmd knockdown tenocytes stimulated with mechanical stretching (Figure 2C,D). As expected, the migration of tenocytes was also significantly reduced in Tnmd knockdown cells exposed to mechanical stretching (Figure 2E,F). These findings suggest that tenocytes translated force into Tnmd signals that promote the migration of tenocytes.

### 2.3. Stretch-Reinforced Actin Stress Fibers Increase Tnmd Expression and Tenocyte Migration 

The actin cytoskeleton is essential for rapid mechanical signals transmission within the cells. To explore the possible role of actin cytoskeleton in the stretch-induced Tnmd and tenocyte migration, we evaluated F-actin cytoskeleton organization in response to mechanical stretching. Compared with the unstretched group, stretched-tenocytes exhibited thickening actin stress fibers (Figure 3A,B). To further investigate the effect of actin stress fibers on stretch-induced Tnmd expression and tenocyte migration, the expression of Tnmd and tenocyte migration was detected after pre-incubation with actin-depolymerizing drug Lat A (0.5 μM), which effectively disrupts the formation of actin stress fibers (Figure 3C,D). We found that the tenocytes exposed to stretching in the presence of Lat A displayed decreased Tnmd expression compared to the stretched tenocytes without Lat A. The transwell assay showed that tenocyte migration was increased in cells subjected to stretching, whereas cells stretched in the presence of Lat A displayed a significant decrease in the tenocyte migration (Figure 3F,G). Moreover, considering that the Rho-associated coiled–coil kinase (ROCK) is a specific factor of actin stress fibers, we evaluated the effect of ROCK inhibitor Y27632 on stretching-increased Tnmd expression and tenocyte migration. The results showed that inhibition of actin stress fibers using Y27632 abrogates stretching-induced Tnmd expression and tenocyte migration (Appendix A). These results suggest that actin stress fibers are important for induction of Tnmd and tenocyte migration in stretched cells.

### 2.4. Stretch-Induced Chromatin Decondensation Increases Tnmd Expression and Tenocyte Migration

To determine whether chromatin organization is mediated by mechanical stimuli, we assessed the state of chromatin condensation in loaded cells. Considering that in situ DNase I sensitivity assays can be used to assess the degree of chromatin condensation [16], we adapted the DNase I sensitivity assay to investigate the effect of mechanical loading on chromatin condensation. The results showed that the nuclear area of cells subjected to stretching without DNase I were indistinguishable from those of the unstretched group (Figure 4A,B). However, DNase I treatment resulted in smaller nuclear areas in the loaded cells stimulated than in the unloaded cells (Figure 4A,B), indicating that mechanical stretching leads to chromatin decondensation.

We next determined whether chromatin decondensation is responsible for stretch-induced Tnmd expression and migration. Two specific agents—histone acetyltransferases inhibitor (ANA) and demethylases inhibitor GSK-J4—were used to inhibit chromatin decondensation. The results showed that treatment of tenocytes with ANA or GSK-J4 resulted in increased chromatin condensation in loaded cells compared to the loaded cells in the absence of an inhibitor (Figure 4C,D). To determine the effect of condensed chromatin on stretch-induced Tnmd expression and tenocyte migration, tenocytes were preincubated with ANA or GSK-J4 for 1 h and then stimulated with stretching. The results showed that tenocytes exposed to stretching in the presence of chromatin decondensation inhibitor ANA and GSK-J4 had a lower expression of *Tnmd* at the mRNA level (Figure 4E). Moreover, we found that, compared with stretched tenocytes, Tnmd protein expression and tenocyte migration were abrogated in ANA-treated loaded cells or GSK-J4-treated loaded cells (Figure 4F–H, respectively). These results suggest that chromatin condensation resulting from histone acetylation and methylation might contribute to stretched-induced Tnmd expression and tenocyte migration.

## 3. Discussion

Mechanical forces are crucial for tendon development, maturation and functional maintenance [17]. Tenocytes are responsible for sending and translating biomechanical signals into biochemical signals. The translated biochemical signals alter tendon-related gene expression to induce cellular functions that enhance tissue maturation, remodeling, and/or repair [18,19]. Understanding the molecular and cellular events underlying mechanically induced tendon responsiveness could provide information on potential target sites for tendon repair therapy. Tnmd is the best-known marker of the mature tendon [20]. Maintaining the high expression of Tnmd plays an important role in tendon development, functional maintenance, and repair [21]. Mice undergoing moderate mechanical loading displayed a stronger expression of Tnmd [10]. The mechanical loading of TSPCs and tenocytes at a 4% and 8% stretch significantly stimulated Tnmd expression [10]. The *Tnmd* gene was decreased in cultivated tenocytes with passages, upon a 5% axial cyclic strain, and its transcription was sharply increased [22]. Furthermore, the absence of tensile strain causes a reduced expression of Tnmd in tendons in vivo and in tendon fibroblasts in vitro [23]. In our study, Tnmd expression was more reduced in cultured tenocytes in vitro than in tissue, suggesting that its downregulation is a consequence of a lack of mechanical stimuli in vitro. Stretched tenocytes using a mechanical loading system that mimics the in vivo tensile mechanical loading on tenocytes showed that Tnmd upregulates with a 10% cyclic stretching strain. These results suggest that mechanical tension is crucial for the initiation and maintenance of Tnmd in tendon tissues and cells.

Tnmd showed a lower expression in early tendon healing and a higher level in late stages, such that tendons gradually restored their function of mechanical loading [24,25]. Meanwhile, the numbers of tenocytes were increased in the injury area during late-stage healing, suggesting that Tnmd might be involved in the migration or proliferation during healing. Here, we showed that migratory tenocytes were increased in stretching-stimulated tenocytes, and knockdown of Tnmd reduced tenocyte migration in the absence and/or presence of mechanical stretching, suggesting that Tnmd directly mediates the migration of cells. These results suggest that Tnmd could be a potential target for improvement in tendon healing. Targeting the Tnmd gene or proper exercise training to induce high expressions of Tnmd after injury might induce the cells to migrate quickly to the injury area. Studies still need to further verify this in clinical experiments.

It has been reported that mechanical stretching induces the actin stress fibers thickening of fibroblasts plated on a flexible substrate [26]. Moreover, the polymerization state of actin regulates the cytoplasmic to the nuclear localization of various transcription factors that regulate gene expression [12]. In addition, Zhu et al. reported that forcing tenocytes into an elongated shape using micropatterning techniques leads to the formation of actin stress fibers, accompanied by upregulation of the *Tnmd* gene. Furthermore, it has been reported that cytoskeleton tension is reduced by enforced elongated morphology. Interestingly, the disruption of actin stress fibers formation treatment with Y27632 could significantly increase the expression of *Tnmd* in the spread tenocytes, albeit without a significant difference in the *Tnmd* level in elongated tenocytes [27]. By contrast, our results showed that the disturbance of actin stress fiber formation with Lat A could reduce the Tnmd expression of tenocytes. The two results are different because micropatterning-enforced elongated tenocytes already had a low cytoskeleton tension induced by actin stress fibers along the long axis of cells, so Y27632 was not able to further change cytoskeleton tension. However, in our research, compared with a normal cultured and unstretched group, treatment with Lat A destroyed the structure of actin stress fibers, which reduced the cytoskeleton tension loading of tenocytes. These findings indicate that Tnmd expression might be associated with actin cytoskeleton tension. Our results show that Tnmd expression is reduced in Lat-A-treated tenocytes in the presence of stretching compared with normal cultured and stretched tenocytes. These results suggest that Tnmd expression is closely associated with the tension from the actin cytoskeleton remodeling.

Many studies in recent years have highlighted the importance of chromatin remodeling within the nucleus as crucial regulators of gene expression [28]. For example, cell geometry-induced actin remodeling has been shown to result in chromatin condensation, which can lead to transcriptional silencing [29]. Reducing actomyosin contractility using an actin polymerization inhibitor results in the shuttling of histonedeacetylases (HDACs) to the nucleus, which leads to enhanced chromatin condensation [15,30]. Force applied to the cell surface can directly stretch chromatin, and that stretching can activate local gene transcription. In our study, inhibitors (ANA or GSK-J4) were used to treat tenocytes to assess the exact effect of the chromatin condensation state on Tnmd expression and tenocyte migration induced by stretching, and we found that inhibitors do not affect the protein expression of *Tnmd* in unstretched cells. However, compared with stretched tenocytes, *Tnmd* expression was abated in stretched tenocytes in the presence of inhibitors, accompanied by a significant inhibition of chromatin decondensation, suggesting that chromatin decondensation induced by stretching contributes to Tnmd levels. Viala et al. reported that the condensed chromatin does not affect the initiation of transcription. Chromatin decondensation triggers increased transcription [29]. These results suggest that mechanical stretching upregulates Tnmd expression via increased chromatin decondensation. Interestingly, our results show that cell migration was reduced in the presence of ANA or GSK-J4, compared with the unstretched tenocytes, suggesting that chromatin condensation regulates tenocyte migration via the regulation of other gene transcriptions. These results suggested that the decondensation of chromatin induced by mechanical stretching contributes to the expression of Tnmd and tenocyte migration. However, determining whether the chromatin organization changes that occur in response to stretching directly regulate tenocyte migration or occur by altering Tnmd expression will require further research.

We have provided evidence that Tnmd positively regulates tenocyte migration and that Tnmd expression can be better maintained by mechanical stretching. Moreover, this study reveals that mechanical stretching signals are transmitted via actin stress fibers in the nucleus to condensed chromatin that promotes the expression of Tnmd, which might then increase tenocyte migration. However, there are some limitations here: First, even though we demonstrated that the inhibition of actin stress fibers using Lat A treatment reduced stretch-induced Tnmd expression and cell migration, it is worth mentioning that Lat A function is not specifically actin fibers, and more specific techniques are required. For example, it will be informative to interfere with the stress fibers formation of function by the knockdown of proteins such as α-actinin, Alix and paladin. Second, the molecular signal transduction pathway involved in the above process is still not clear and needs to be further studied.

## 4. Materials and Methods

### 4.1. Cell Isolation and Culture 

All the animal experimental procedures were performed according to ethical standards and national and international standards, and they were approved by the Chongqing Science and Technology Commission, Chongqing, China. All Sprague-Dawley (SD) *rats*, regardless of their gender, weighing 80–120 g, provided by Chongqing Medical University, were used. The isolation of primary tenocytes refers to our previous study [31]. The tenocytes were cultured in low-glucose Dulbecco’s modified Eagle’s medium (DMEM; Gbico, MA, USA) supplemented with 10% fetal bovine serum (Hyclone, Logan, UT, USA), 2 mM glutamine, penicillin (100 U/mL) and streptomycin (100 mg/mL). The cultivating medium was changed every two days. Tenocytes were used between Passage 1 to 4.

### 4.2. Total RNA Extraction and Quantitative RT-PCR

Total RNA was extracted with RNA Extraction Kit (TaKaRa, Kyoto, Japan) Reverse transcription was performed using the PrimeScript^TM^ Reagent Kit (TaKaRa, Kyoto, Japan) according to the manufacturer’s instructions. Real-time quantitative PCR analyses were implemented using TB Green Premix Ex Taq^TM^ II (TaKaRa, Kyoto, Japan). Expression was analyzed using the CFX Manager Software 3.1 (Bio-Rad, Hercules, CA, USA). The expression was calculated relative to GAPDH. The sequences of the primers are shown in Appendix A.

### 4.3. Western Blotting

Stimulated cells were washed with cold PBS, lysed in RIPA, and supplemented with PMSF (Beyotime Biotechology, Shanghai, China). The lysates were modified to similar concentrations, mixed with a 6× sample loading buffer, and boiled for 10 min. The lysates were resolved in a 10% SDS-polyacrylamide gel and were electrotransferred to Polyvinylidene fluoride membranes (Millipore, Billerica, MA, USA). After blocking with 5% nonfat milk in Tris-buffered saline (TBS) with 0.05% Tween 20 (TBST), the membranes were incubated with primary antibodies (Tnmd, Abcam, ab203676, Cambridge, UK) (GAPDH; ZenBio, Chengdu, China) overnight at 4 °C. The membranes were incubated with an appropriate secondary antibody (Beyotime Biotech, Shanghai, China) for 1 h, followed by detection using an ECL blotting analysis system (Bio-OI, Ghangzhou, China).

### 4.4. Uniaxial Cyclic Stretch 

The cyclic stretching device (Model ST-140; STREX Co., Ltd., Osaka, Japan) was used to generate stretching loading to tenocytes in vitro. Cells (1× 10^5^ cells) were seeded onto elastic silicone chambers (2 cm × 2 cm) that had been pre-coated with 5 μg/mL rat tail type I collagen (Hangzhou shengyou Biotechnology Co., Zhejiang, China) and grown to confluence overnight. Cells were subjected to uniaxial cyclic stretching (10%, 0.5 Hz, 1 h). For stretching experiments with inhibitors (0.5 μM Lat A, 10 μM ANA, 10 μM GSK-J4 and 10 μM Y27632; MedChemExpress, Monmouth Junction, NJ, USA), cells were preincubated for 1 h prior to stretching, and the stretching was performed in media with inhibitors added. As a control, static cells were cultured in a chamber under the same conditions without any stretching.

### 4.5. Transwell Migration Assay

Tenocyte migration was assayed using Transwell chambers (8 μM, Millipore, Burlington, MA, USA). 1.5 × 10^4^ tenocytes in 100 µL of serum-free medium were placed in the upper room, and 600 μL of complete medium (10% FBS) were added to the bottom well. After incubation for 12 h, the upper room was wiped gently with a cotton swab in order to remove the non-migrated tenocytes. The cells on the lower filters were fixed in 4% formaldehyde for 15 min and stained with crystal violet (0.05%) for 30 min, then washed in PBS, and dried. Six fields on the underside of the chamber were selected randomly, and the numbers of tenocytes therein were counted and averaged for the drug treatment during the migration assay. The drugs were added to both the top and bottom wells.

### 4.6. Immunofluorescence

Cells were fixed with a 4% paraformaldehyde solution for 15 min and washed for 5 min three times with PBS. The cells were then permeabilized and blocked with 0.2 % Triton X-100/PBS for 3 min, and then rinsed by PBS for 5 min three times. They were then blocked using 1% BSA/PBS for 1 h. Cells were incubated with primary antibodies anti-tenomodulin antibodies (dilution, 1: 200; Abcam, Cambridge, MA, USA) overnight at 4 °C, followed by Alexa Fluor 555-labeled secondary antibodies (dilution,1:500; Beyotime, Shanghai, China). Actin filaments and DNA were stained with Phalloidin-Tetramethylrhodamine Conjugate (Sigma-Aldrich, Darmstadt, Germany), and 500 nM DAPI (Beyotime, Shanghai, China), respectively. Images were captured with a Leica fluorescence microscope (Leica, Solms, Germany).

### 4.7. Stress Fiber Thickening Analysis

Stress fiber thickening (SFTI) was analyzed using the model of an erosion rank filter. In brief, the picture was opened in Image J and inverted, and an applied erosion filter yielded a brightness decay that indicated the thickness of actin bundles. The decay constant was termed SFTI. Stress fibers/tenocytes in >12 fields were analyzed and averaged (*n* > 300).

### 4.8. Construction of Lentiviral Vectors and Lentiviral Transduction

For Tnmd knockdown, short hairpin oligonucleotides directed against Tnmd were designed. The shRNA sequences were shown in Appendix A. The sequences of the shRNAs were cloned in the vector pLKO.1-puro (Addgene, Cambridge, MA, USA). For lentivirus production, the helper plasmid pSPAX2 and PMD2.G (Invitrogen, CA, USA) were transfected into 293FT cells with lipofectamine 3000 (Invitrogen, California, CA, USA). The culture supernatants were collected at 72 h and then subjected to 0.45-μm filters (Millipore, Billerica, MA, USA). The tenocytes were infected with the lentivirus in the presence of 10 μg/mL polybrene (Sigma Aldrich, Darmstadt, Germany). After 24 h, the cells were washed with PBS and cultured in a selection medium including 2 μg/mL puromycin (Beyotime Biotech, Shanghai, China). After incubation for 7–15 days, cells were seeded onto precoated (5 μg/mL rat tail type I collagen) silicone membranes and grown to confluence overnight. Cells were then subjected to mechanical stretching.

### 4.9. DNase I Sensitivity Assay

For the in situ DNase I sensitivity assays, we followed a previously described protocol [32]. In brief, cells were washed with a CSK buffer (100 mM NaCl, 0.3 M sucrose, 10 mM PIPES, 3 mM MgCl2, PH 6.8) supplemented with 0.2% Triton X-100 and a protease inhibitor cocktail (Sigma-Aldrich, Darmstadt, Germany) for 5 min. The samples were then incubated in the CSK buffer with the protease inhibitor cocktail, 100 U/mL DNase I (Sigma-Aldrich, Darmstadt, Germany), and 0.2 % Triton X-100 for 20 min at RT. The DNA was stained using DAPI for 10 min at RT. The sample were fixed in methanol for 5 min at RT and washed with the CSK buffer. Images were captured using a fluorescence microscope (Leica, Solms, Germany). Nuclear areas of cells were measured using Image J software 1.4.3.67 (NIH, Bethesda, MD, USA).

### 4.10. Statistical Analysis

Graphs showed means with ±SD, *p* Values were analyzed by one-way analysis of variance followed by the Student’s t-test. *p* < 0.05 was considered statistically significant.

## Figures and Tables

**Figure 1 ijms-22-04928-f001:**
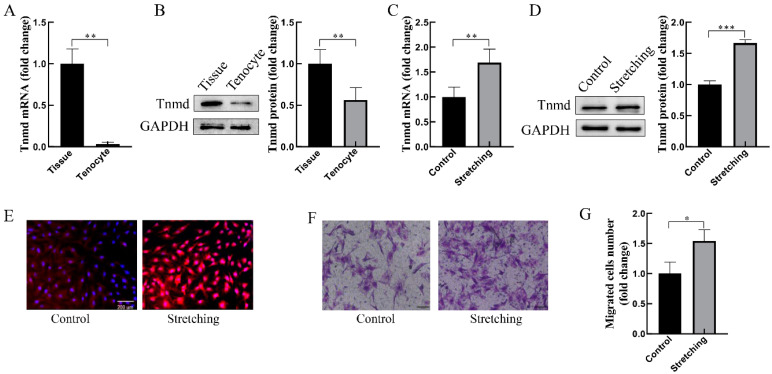
Stretching upregulated Tnmd expression and cell migration of cultured tenocytes. (**A**) RT-qPCR detection of *Tnmd* mRNA expression level. (**B**) Western-blotting (WB) detection of the Tnmd protein level in tendon tissue and cultured tenocytes. (**C**–**E**) The gene and protein expression of Tnmd was analyzed in loaded tenocytes by RT-qPCR, WB and immunofluorescence (scale bar, 200 μm), respectively. (**F**) Migratory tenocytes were assayed by a transwell chamber (scale bar, 100 μm). (**G**) The migration rate was quantized. GAPDH (glyceraldehyde-3-phosphate dehydrogenase) was used as a loading control protein. The graph shows the mean with SD; *n* = 4, * *p* < 0.05; ** *p* < 0.01; *** *p* < 0.001.

**Figure 2 ijms-22-04928-f002:**
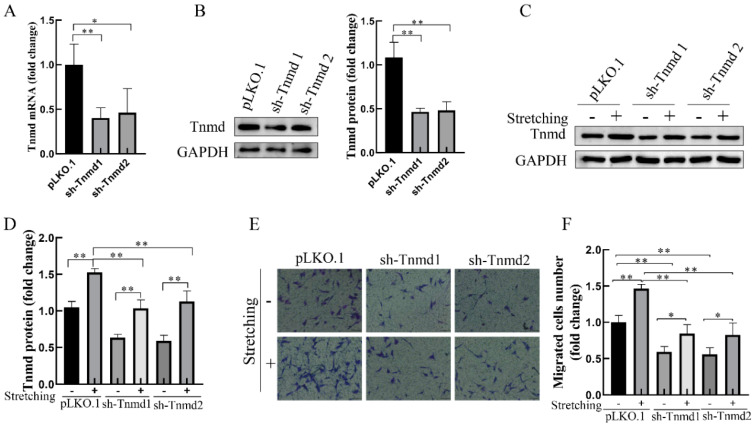
Stretching enhanced tenocyte migration via increased Tnmd expression. (**A**,**B**) RT-qPCR and WB detection of Tnmd mRNA and protein expression. The tenocytes were denoted as “pLKO.1” and were infected with lentivirus-shcontrol. Tenocytes were denoted as “sh-Tnmd” and interfered with lentivirus targeting Tnmd. Two independent Tnmd shRNAs, denoted as sh-Tnmd1 and sh-Tnmd2, were utilized. (**C**,**D**) The protein expression of Tnmd and GAPDH (loading control) were detected by WB. (**E**) Migratory tenocytes were detected by a transwell assay (scale bar, 100 μm). (**F**) The migration was counted and quantized. The graph shows the is mean with SD; *n* = 4, * *p* < 0.05; ** *p* < 0.01.

**Figure 3 ijms-22-04928-f003:**
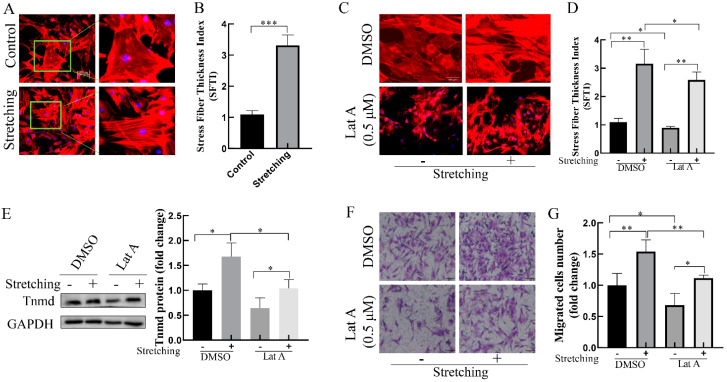
Thickening of actin stress fibers contributes to Tnmd expression and tenocyte migration in loaded tenocytes (**A**,**C**) The F-actin organization was detected via phalloidin staining, nuclei were stained with DAPI (blue) (scale bar, 100 μm). The right column is a partial enlarged view of the left original image green border (in Figure 1A). (**B**,**D**) F-actin cytoskeleton remodeling was evaluated and quantized with stress fiber thickening (SFTI). (**E**) WB was used to analyze the expression of Tnmd protein. (**F**) Migratory tenocytes were detected by a transwell assay (scale bar, 100 μm). (**G**) The migration was counted and quantized. The graph shows the mean with SD; *n* = 4, * *p* < 0.05; ** *p* < 0.01; *** *p* < 0.001.

**Figure 4 ijms-22-04928-f004:**
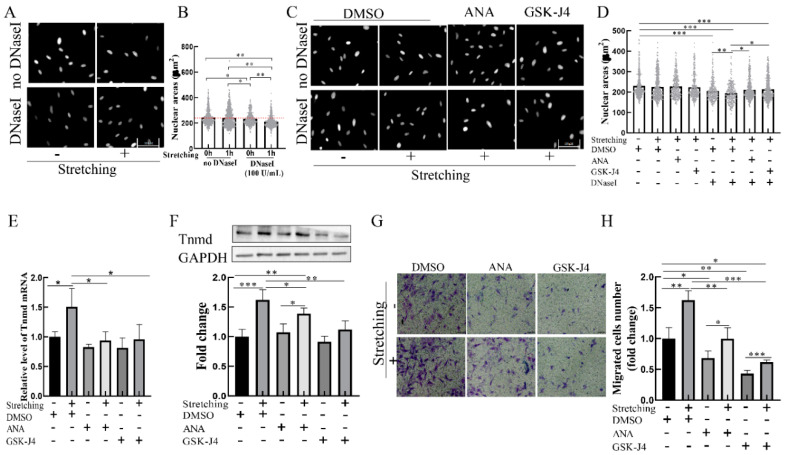
Decondensed chromatin contributes to Tnmd expression and tenocyte migration in loaded tenocytes. (**A**,**C**) Chromatin condensation was detected by in situ DNase I sensitivity (scale bar, 100. μm). (**B**,**D**) The degree of chromatin condensation was evaluated and quantized with nuclear areas. (**E**) RT-qPCR detection of *Tnmd* mRNA expression. (**F**) WB was used to analyze the expression of Tnmd protein. (**G**) Migratory tenocytes were detected by a transwell assay (scale bar, 100 μm). (**H**) The migration was counted and quantized. The graph shows the mean with SD; *n* = 4, * *p* < 0.05; ** *p* < 0.01; *** *p* < 0.001.

## Data Availability

All data is contained within the article.

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
