# Peer review of "Stretch-Induced Tenomodulin Expression Promotes Tenocyte Migration via F-Actin and Chromatin Remodeling"

_ijms, 2021, doi:10.3390/ijms22094928_

Round 1
Reviewer 1 Report
The authors showed that Tenomodulin expression in tenocyte is up-regulated by chromatin decondensation caused by actin filament network expansion when tension is applied to the cell, which is involved in the cell migration.
The results are easy to follow and obvious. I believe this manuscript is worth publishing.
I would like to point out some small things to be revised before publication.
1: Some abbreviations are not explained when they first appear, “GAPDH” in Figure 1, “sh-Tnmod1”, “sh-Tnmod2”, “PLKO.1” in Figure 2 and “HDAC” (page 7, line 236). And the authors should explain in the results section that PLKO.1 and GAPDH were used as controls.
2: Page 7, lines 223-226: “The two results are different because micropatterning-en-forced elongated tenocytes already had a low cytoskeleton tension induced by actin stress fibers along the long axis of cells and thus Y27632 was not able to further change tension loading of tenocytes to regulate Tnmd expression.” It is difficult to follow why the micropatterning techniques reduced the cytoskeleton tension. More explanation is required.
Author Response
Thanks greatly for the critical review and valuable comments. Please find the attachment.

Reviewer 2 Report
The manuscript by Xu et al. describes some aspects of the molecular mechanism that is responsible for tenocyte migration in response to mechanical forces. More specifically, the authors studied the involvement of Tenomodulin (Tnmd) in this process. The authors show that exposure of tenocytes to stretching forces leads to an increase in Tnmd expression level at the RNA and protein levels. By knock down experiments the authors show that Tnmd is important for tenocyte migration both in non-pre-treated cells and in cells that were exposed to mechanical stretch before the migration assay. Thus, the authors suggest the Tnmd is important for the mechanotransduction mechanism that leads to increased migration rate of tenocytes. In search for a mechanistic insight the authors show that stress fiber formation in response to stretch is important for the increase in Tnmd levels. The authors also try to link stretching forces to changes in chromatin organization. Towards this direction the authors show that stretching forces induce chromatin decondensation by DNase I sensitivity assay and that interference with chromatin decondensation reduces Tnmd levels and the cellular migration rate.
Previous reports already showed that Tnmd is a mechanosensitive gene. For example, in Int J Mol Sci. 2020 Feb 6;21(3):1082. doi: 10.3390/ijms21031082 it was shown that Tnmd protein levels are increased in response to stretching forces. It was also reported before that Tnmd is important for tenocytes migration (Cell Death Dis. 2017 Oct 12;8(10):e3116. doi: 10.1038/cddis.2017.510). Thus, the first half of the manuscript (figures 1-2) is mainly a re-establishment of known features of tenocytes. The main novelty of the manuscript is the mechanistic data presented in figures 3 & 4. However, this data is partial and not coherent. In figure 3 the authors show that stress fiber formation is important for increased Tnmd levels and migration rate in response stretching forces. Latrunculin A interferes with actin polymerization in general rather than specifically with stress fibers. It will be informative to interfere with stress fibers formation or function in a more specific way by knock down of proteins such as α-actinin, Alix or paladin. In addition, if stress fibers are studied, it will be interesting to show what is the signal transduction pathway from stress fibers to Tnmd transcription.
In figure 4 the analysis is preliminary. The finding that stretching forces lead to chromatin decondensation was reported before (ref. 15). In panels 4E-G results of non-stretched cells with the inhibitors should be presented. ANA and GSK-J4 should lead to a general reduction in transcription since ANA reduces histone acetylation and GSK-J4 increases H3K27me3 levels. It is not clear if the effects are really Tnmd specific or more general: determining the overall transcription rate should and the RNA levels of Tnmd should be done. In addition, it is important to overexpress Tnmd in ANA and GSK-J4 treated-cells to test if the effects of these drugs are due to alterations in global chromatin organization or specific regulation of Tnmd.
Additional points for improvement
-Tnmd mRNA levels: there is no information regarding any reference gene. Please add this data.
-Western blot analyses: in general, the presented blots are over-exposed. In Fig. 3E, Tnmd levels in LatA treated cells look similar to the levels in the control cells (lanes no. 1 and 3), while in the bar graph there is a difference between the two points. Please clarify.
-Transwell migration: Please indicate the diameter of the pores.
-Stress fiber thickening analysis: It is not clear how the analysis was done. Please explain how the analysis was done.
-Lat A treatment: Lat A concentration was not kept constant in the text the concentration is 0.25uM, while in some panels in figure 3 it is 50uM. The same concentration of Lat A should be used in all assays to be able to compare them to each other.
-The English should be edited.
Author Response

(The authors gave the same response as above.)

Round 2
Reviewer 2 Report
The authors corrections answer only a part of the concerns I raised before. Two major points were not addressed:
- In studying the link between stress fibers and Tnmd levels (Fig. 3) the authors claim that stress fibers are important for induction of Tnmd in stretched cells. As I wrote previously, the presented data is not enough to support this claim. I asked the authors to interfere with stress fibers function/formation in a more specific way than Lat A, but they did not do anything like that. To be more specific the authors should knock down stress fiber-specific factors or to inhibit such factors. ROCK, which was inhibited in the reference the authors presented in their rebuttal letter is an excellent candidate.
- Regarding the link between chromatin decondensation, Tnmd levels and the cellular migration rate (Fig. 4), in my previous review I asked for the analysis of Tnmd RNA levels in the presence of ANA and GSK-J4, but it was not done. It is important since the discussion on the effect of chromatin condensation level on Tnmd levels is at the transcriptional level. I also asked to over-express Tnmd in ANA and GSK-J4 to check if the effects of the drugs are Tnmd-specific or more general. This analysis was not added.
I am sorry but according to the journal policy the requirement of additional experiments results in rejection of the manuscript.
